# Effects of Virtual Reality-Based Rehabilitation on Burned Hands: A Prospective, Randomized, Single-Blind Study

**DOI:** 10.3390/jcm9030731

**Published:** 2020-03-09

**Authors:** So Young Joo, Yoon Soo Cho, Seung Yeol Lee, Hyun Seok, Cheong Hoon Seo

**Affiliations:** 1Department of Rehabilitation Medicine, Hangang Sacred Heart Hospital, College of Medicine Hallym University, Seoul 07247, Korea; anyany98@naver.com (S.Y.J.); yscho@hallym.or.kr (Y.S.C.); 2Department of Physical Medicine and Rehabilitation, College of Medicine, Soonchunhyang University Hospital, Bucheon 14584, Korea; shouletz@gmail.com (S.Y.L.); 50503@schmc.ac.kr (H.S.)

**Keywords:** virtual reality, burn, hand

## Abstract

Hands are the most frequent burn injury sites. Appropriate rehabilitation is essential to ensure good functional recovery. Virtual reality (VR)-based rehabilitation has proven to be beneficial for the functional recovery of the upper extremities. We investigated and compared VR-based rehabilitation with conventional rehabilitation (CON) in patients with burned hands. Fifty-seven patients were randomized into a VR or CON group. Each intervention was applied to the affected hand for four weeks, and clinical and functional variables were evaluated. Hand function was evaluated before intervention and four weeks after intervention using the Jebsen-Taylor hand function test (JTT), Grasp and Pinch Power Test, Purdue Pegboard test (PPT), and Michigan Hand Outcomes Questionnaire (MHQ). The JTT scores for picking up small objects and the MHQ scores for hand function, functional ADL, work, pain, aesthetics, and patient satisfaction were significantly higher in the VR group than in the CON group (*p* < 0.05). The results suggested that VR-based rehabilitation is likely to be as effective as conventional rehabilitation for recovering function in a burned hand. VR-based rehabilitation may be considered as a treatment option for patients with burned hands.

## 1. Introduction

The distal parts of the upper extremities are the most common sites of thermal injury. The hand is one of the three most common joints of scar contracture deformity following a burn injury [1,2]. Some studies have reported that the total body surface area grafted and injury to the right dominant hand are some of the predictors of hand contracture [1]. The loss of hand function after a burn can have a crucial effect on the activities of daily living (ADL). Some reports have noted that two-thirds of the patients with deep hand burns change their profession because of their hand functional disorder [3]. Functionally, hand contractures may affect one’s ability to perform ADL, such as dressing, eating, and grooming as well as fine motor tasks such as typing, writing, and occupational activities. Common complications after a burn to the hand include joint deformities, sensory impairment, scar contracture, and postburn edema [3].

Appropriate rehabilitation is important to ensure that good functional recovery is achieved [4,5]. Burned hands are usually treated and managed by a multidisciplinary team at a burn center to conservatively manage hypertrophic scars and soft tissue contractures. Rehabilitation of the burned hand should be initiated in the acute stages by means of individualized positioning, splinting, and exercise for improving functional activity [6,7]. Frequent exercise throughout the day is more beneficial than one session of intensive exercise [8,9]. Repeated range of motion (ROM) exercises are helpful in decreasing edema and conditioning the tissue. Despite rehabilitation of the burned hand, impaired hand function may remain [10]. Many interventions have been developed and tested for patients with burns; however, the strategies of hand rehabilitation remain controversial [11,12]. 

Recent studies have recommended repetitive exercises using virtual reality (VR) for functional recovery from upper extremity disorders [13,14] because the effectiveness of task-specific training improved when tasks were ordered in a random practice sequence using repetition and positive feedback. Many types of VR-based rehabilitation apparatus, from commercial video game equipment to robotics, are currently being developed and used. The RAPAEL Smart Glove^TM^ (Neofect, Yong-in, Korea) is a VR rehabilitation tool designed for the distal upper extremities [15]. VR is an interactive and enjoyable intervention that creates a virtual rehabilitation scene in which the intensity of practice can be systemically manipulated [16]. Recently, VR has been used in burn studies mainly for pain reduction during acute burn dressing [17,18]. However, few studies have used VR systems for therapeutic purposes and functional assessments in burn patients. This study aimed to evaluate the effects of VR-based rehabilitation on burned hands and compare the results to those of matched conventional intervention (CON) rehabilitation in patients with burns. 

## 2. Materials and Methods

The present study was a single-blind, randomized controlled trial. We recruited 57 (54 men and 3 women) patients from the Department of Rehabilitation Medicine at Hangang Sacred Heart Hospital in Korea between June and October 2019. Our study was registered on ClinicalTrials (NCT03865641). The study was registered at the Ethics Committee of the Hangang Sacred Heart Hospital (2014-081). Patients provided written informed consent. All participants’ burn scars had re-epithelialized after split-thickness skin graft (STSG). To evaluate the effectiveness of VR rehabilitation accurately, the comparison was limited to the right hand and wrist. We included patients aged ≥18 years with a deep, partial-thickness (second-degree) burn or a full thickness (third-degree) burn with the involvement of >50% of the body surface area of the dominant right hand, with joint contracture (hand and wrist). In addition, these patients were transferred to the rehabilitation department after acute burn treatment and less than 6 months since the onset of the burn injury (Figure 1). 

The study excluded patients who had fourth-degree burns (involving muscles, tendons, and bone injuries), musculoskeletal diseases (fracture, amputation, rheumatoid arthritis, and degenerative joint diseases) involving the burned hand, neurological diseases (such as peripheral nerve disorders), preexisting physical and psychologic disability (severe aphasia and cognitive impairment that could influence the intervention), or severe pain impeding hand rehabilitation. Numbers were assigned to 66 burn patients according to the order of admission who satisfied all the aforementioned criteria. A computer program was used to randomly divide them into the VR group (*n* = 32) or the CON group (*n* = 34). One patient in the VR group completed 17 sessions before dropping out due to scheduling conflict. Two patients in the CON group were forced to drop out before completing the study, including one who completed 11 sessions before dropping out due to unrelated medical issues and one who completed 18 sessions before dropping out due to a personal scheduling conflict. Three patients in the VR group and three patients in the CON group dropped out of the study because they recovered function in the injured hand after 20 sessions and could return to their daily life; therefore, they did not want to undergo serial evaluations and did not visit the outpatient clinic. Data from these subjects were not included in the analyses. Thus, patients were divided into two groups (28 patients in the VR group and 29 patients in the CON group) (Figure 2). All patients received a four-week intervention (20 sessions for 60 min per day). The VR group received 30-min standard therapy and 30-min VR-based rehabilitation. The CON group received 60-min standard therapy. The VR and CON interventions focused on the burned hand. Every effort was made to match the exercise rehabilitation in both groups and to adapt the level of difficulty to each patient’s performance. The intervention frequency and duration did not differ between the VR and CON groups. The rehabilitation was supervised by three occupational therapists who had experience with VR rehabilitation tools and conventional therapy. Therefore, all factors, except for the use of the VR system, were consistent between the 2 groups.

We applied the RAPAEL Smart Glove^TM^ (Neofect, Yong-in, Korea) system, which combined the use of an exoskeleton type glove and the VR system (Figure 3). This tool can be operated only through active movement and not through passive movement. The software can be used to visualize the virtual hands in the VR tool according to the data gathered by a glove-shaped sensor device. The training programs demanded the following volitional movements: forearm pronation/supination, wrist flexion/extension, wrist radial/ulnar deviation, and finger flexion/extension (Figure 4). Visual and audio feedback informed the patients of success or failure. The patients were required to accurately complete a task in time and with proper power to obtain high scores. The standard therapy comprised range of motion (ROM) exercises for the burned hand, strengthening exercises of the upper extremities using tabletop activities, manual lymphatic drainage, and desensitizing sensory stimulation of hypertrophic scars. 

A squeeze dynamometer (Lafayette Instrument, Lafayette, IN, USA) was used to measure grip strength, and the Michigan Hand Outcomes Questionnaire (MHQ) [19] was used to assess a patient’s perception of hand function. All MHQ scale scores are based on a scale from 0 to 100. For all scores, except for pain scores, higher scores indicate better hand performance. For the pain scale, lower scores indicate less pain. The Jebsen–Taylor hand function test (JTT) measured the performance speed of standardized tasks [20]. The JTT involves a series of 7 subtests. We used a scoring system in which each subtest score ranged from 0 to 15, and the total score ranged from 0 to 105. The Purdue Pegboard test (PPT) was used to measure fine hand motor functions and dexterity. The scores in the PPT indicate the numbers of pins placed in a board with 2 parallel rows of 25 holes within 30 s. The assembly score in the PPT is assessed for the number of assembled pins, washers, and collars in 60 s. We recorded scores for the right affected hand, both hands, and assembly. The grip strength, JTT, PPT, and MHQ scores were measured immediately before the intervention and after the 4-week intervention. Outcome measurements and data analyses were performed by a trained and blinded outcome assessor who was not involved in the intervention.

Statistical analysis was performed using SPSS, version 23 (International Business Machines (IBM) Corp., Armonk, NY, USA). Parametric data was analyzed using a paired *t*-test after testing for normality. Non-parametric data was analyzed using the Mann–Whitney test and the Wilcoxon signed-rank sum test. To examine the pretreatment homogeneity between the VR and CON groups, the Mann–Whitney test was used for age, time to treatment, Grasp and Pinch Power Test, PPT, all areas of the JTT, and all areas of the MHQ, with a significance level of *p* < 0.05. The changes before and after intervention were compared using the independent *t*-test for tip pinch strength between groups. The changes before and after intervention were compared using the Mann–Whitney test for all parameters except pinch strength, with a significance level of *p* < 0.05. The scores after intervention were compared between the two groups using the Mann–Whitney test for grasp strength, lateral pinch strength test, all areas of the PPT, all areas of the MHQ, and all areas of the JTT. As this was the first study to assess the efficacy of the VR-based rehabilitation in patients with burns, power calculation was based on a previous study, which applied VR-based rehabilitation for the distal upper extremity function in patients with stroke [15]. Accordingly, eighteen patients were required in each group to provide 80% power for efficacy evaluation, setting the α level at 0.05. We calculated that a minimum 52 patients were needed, considering a 30% dropout rate.

## 3. Results

Within six months after a burn injury, 57 patients had received rehabilitative therapy. Before the intervention, there were no differences in demographic and clinical characteristics between the patients in the VR and CON groups (Table 1).

There were significant improvements in the grasp, lateral pinch, and tip pinch strength changes taken before and after intervention in the VR group compared with the changes in the CON group (*p* = 0.03, *p* = 0.002, *p* = 0.03, respectively) (Table 2). There were no significant differences in the grasp, lateral pinch, and tip pinch strength between the two groups after intervention (*p* = 0.13, *p* = 0.06, and *p =* 0.07, respectively) (Table 3). In the JTT, the subtest scores for picking up small objects and simulated feeding in the VR group showed significant changes, compared with the changes in the CON group (*p* < 0.001 and *p =* 0.04) (Table 2). The subtest score for picking up small objects was significantly higher in the VR group than in the CON group after the intervention (*p* = 0.01) (Table 3). There were no significant changes between the groups in the affected hand, both hands, and assembly (*p* = 0.58, *p* = 0.69, and *p =* 0.96, respectively) (Table 2). There were no significant differences in the affected hand, both hands, and assembly between the two groups after intervention (*p* = 0.74, *p* = 0.33, and *p =* 0.41, respectively) (Table 3).

In the MHQ scores, the subtest scores for functional ADL and patient satisfaction in the VR group showed significant changes, compared with the changes in the CON group (*p =* 0.003 and *p* = 0.001) (Table 2). There were significant improvements in the MHQ scores for functional ADL, work, pain, and patient satisfaction differences between the two groups (*p* = 0.005, *p* = 0.04, *p* = 0.002, and *p =* 0.02) (Table 3) No patient experienced adverse events such as skin abrasions or worsening of joint pain during training. In addition, no surgery-related adverse events were reported. 

## 4. Discussion

After the intervention, the subtest scores in the JTT (picking up small objects) and subscale scores for ADL, work, pain, and satisfaction in the MHQ were significantly higher in the VR group than in the CON group. Minimum clinically important changes for scoring system in the JTT and MHQ have not been established in patients with burns. However, results of the JJT and MHQ showed significantly the improvements in VR group [16,21]. Participants were able to engage with the VR system and complete the interventions without premature withdrawal or non-adherence. A previous study has emphasized the importance of robot-assisted rehabilitation in the acute phase after musculoskeletal injuries and reported that it allowed the patient to mobilize the affected arm early by reducing muscle mass loss and improving motor capacity [22]. Immobilization after burn injuries induces loss of muscle strength and flexibility in adjacent joints. Recent studies have emphasized acute rehabilitation in patients with burns [23]. Schwickert et al. reported that VR treatment immediately after orthopedic surgery suppressed pain perception as the patient’s attention was highly focused on task completion [24]. The role of immersive VR in pain reduction has been proven during physical therapy and burn wound care in patients with burns [17,18,25]. These studies found that VR rehabilitation in the acute rehabilitation phase after STSG was well-tolerated by patients with burns.

Importantly, significant changes in multiple outcome measures were observed in the VR group after intervention. Jeffery et al. reported that the VR system can be modified to maximize therapeutic benefits such as improving the ROM of the shoulder and elbow joints after burn injury [26]. The tasks in the VR system include ADLs, such as cooking, cleaning windows, squeezing oranges, and turning over pages. The patients were required to accurately complete each task in time and with proper power, to gain high scores. This provides a feeling of satisfaction in the MHQ that leads to positive reinforcement. A recent randomized controlled study showed that the subtest scores in the JTT were greater in the VR rehabilitation group than in the CON rehabilitation group in patients with hand impairments after stroke [16]. In addition, our study found that VR-based rehabilitation may be more effective in improving hand function as CON rehabilitation, according to the JTT and MHQ scores (pain and satisfaction).

We found that VR rehabilitation was effective according to the MHQ scores for ADL and work. Park et al. reported that upper extremity rehabilitation improved hand function in patients, in addition to elbow and forearm improvement [27]. Functions that include grasp power are associated with proximal upper extremity functions. Previous studies have found that task-specific rehabilitation improved upper extremity functions [28]. Some reports have shown that the effects of VR-based upper extremity rehabilitation had been transferred to untrained tasks [13,16,29]. This study found generalized improvement in unpracticed motor tasks after intervention for ADL tasks. 

Moreover, many reports regarding neurorehabilitation have suggested that for functional recovery in the upper and lower extremities, repetitive intervention using VR is helpful and induces neuroplasticity [14,30,31,32]. Saleh et al. observed the different mechanisms between robot-assisted VR training and conventional functional rehabilitation in patients with chronic stroke using functional magnetic resonance imaging (MRI). Their study concluded that the different neural reorganization was due to the greater visuomotor feedback of the robot-assisted VR system than that of the conventional functional rehabilitation [33]. Further studies are required to evaluate the changes in brain activity or peripheral neuromuscular functions to improve our understanding of the mechanisms of VR rehabilitation after musculoskeletal injury, including burns. This study has limitations that require cautious interpretation of the data, including a relatively small sample size and narrowly focused inclusion criteria. For evaluating the efficacy of VR rehabilitation in patients with burned hand, the hand ROM parameter could be analyzed. However, the hand ROM parameter was not included in the routine hand function evaluations in this study. Further studies are required to compare efficacy in terms of edema reduction and improvement in hand ROM between VR-based and CON rehabilitation and to determine appropriate intervention levels for burned hands. 

## 5. Conclusions

VR-based rehabilitation may be as effective as conventional hand rehabilitation for improving hand power, hand function, and MHQ scores. Burn survivors with acute hand impairment after STSG were able to successfully use the VR system and had significantly better scores on pain, ADL, work and satisfaction scales on the MHQ than the control group. As a result of this study, VR-based intervention is likely to be a clinically useful rehabilitation tool for patients with burned hands. Furthermore, studies that include hand rehabilitation for fourth-degree burns with severe neuromuscular injuries along with treatment intensity, frequency, and VR-based intervention interval studies are needed.

## Figures and Tables

**Figure 1 jcm-09-00731-f001:**
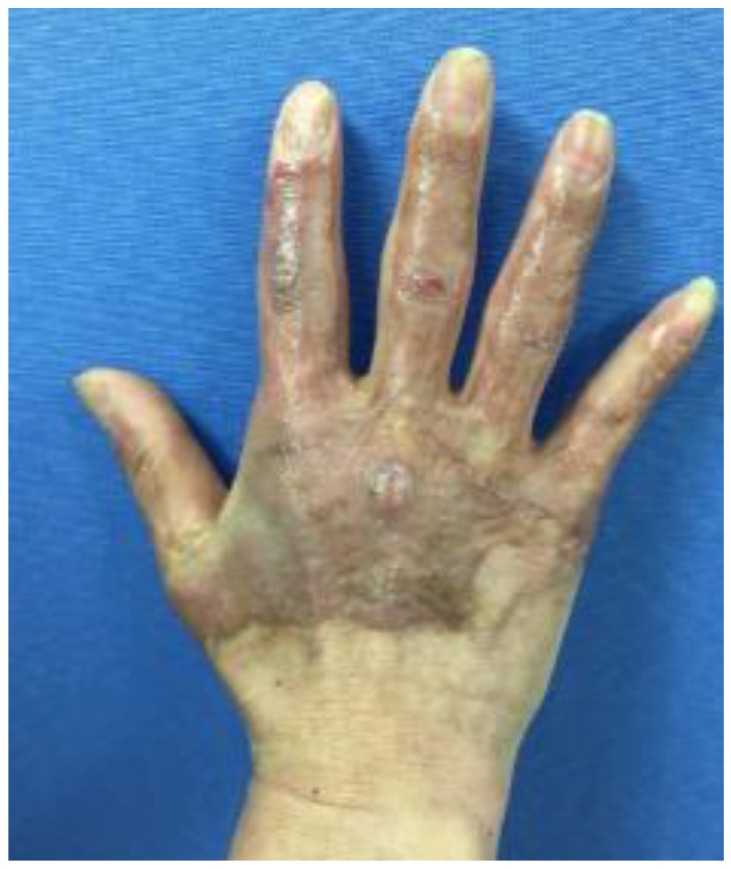
Patient with a deep partial-thickness burn or a full thickness burn with the involvement of >50% of the body surface area of the dominant right hand.

**Figure 2 jcm-09-00731-f002:**
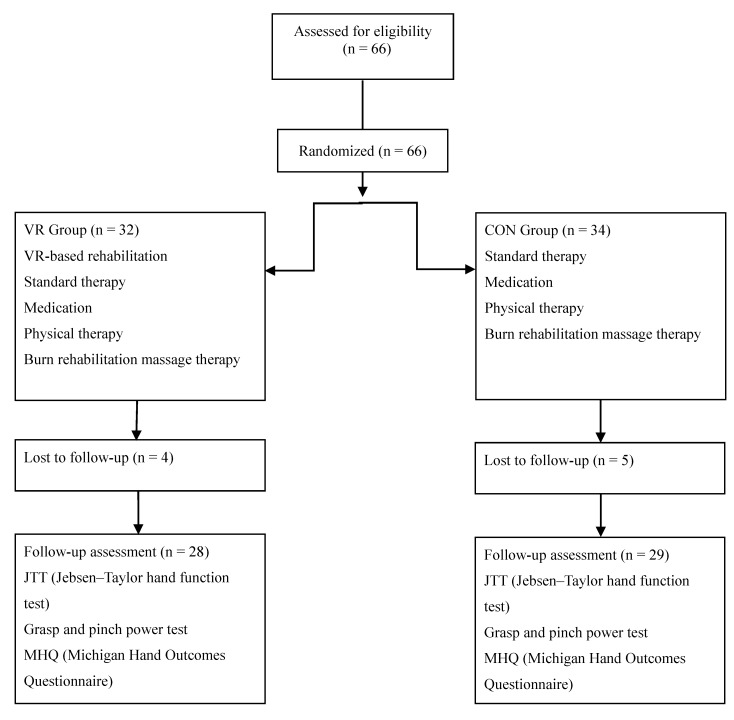
Diagram for subject enrollment, allocation, and follow-up. VR: virtual reality; CON: conventional rehabilitation

**Figure 3 jcm-09-00731-f003:**
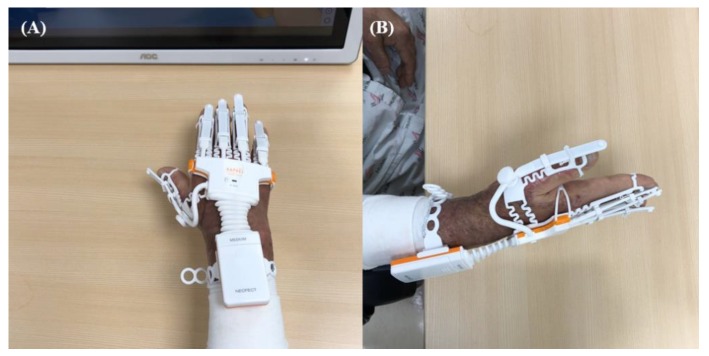
Virtual reality-based rehabilitation system on the burned hand of a study patient: (**A**) antero-posterior view (**B**) lateral view.

**Figure 4 jcm-09-00731-f004:**
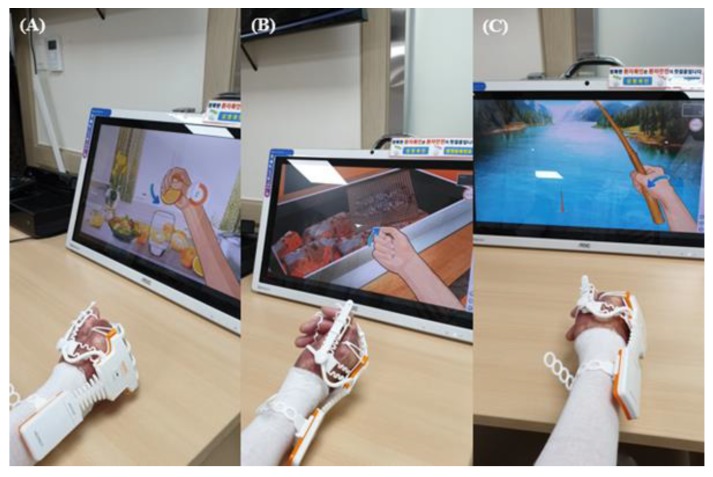
Volitional movements in the virtual reality-based rehabilitation system: (**A**) finger flexion/extension, (**B**) forearm pronation/supination, (**C**) wrist flexion/extension.

**Table 1 jcm-09-00731-t001:** Baseline characteristics of the study patients.

	VR Group (*n* = 28)	CON Group (*n* = 29)	*p*-Value
Male: Female	28:0	26:3	0.24
Age (years)	48.07 ± 8.14	41.69 ± 14.05	0.21
Cause of burn			0.06
Flame burn	18	16	
Electrical burn	2		
Contact burn		6	
Scalding burn	2	4	
Spark burn	6	3	
Time to treatment (days)	74.79 ± 24.15	83.79 ± 46.22	0.98
TBSA (%)	27.71 ± 20.15	27.38 ± 20.65	
Grasp and Pinch Power Test			
Grasp (kg)	4.35 ± 4.73	4.59 ± 3.98	0.53
Lateral Pinch (kg)	3.23 ± 1.38	3.97 ± 2.04	0.05
Tip Pinch (kg)	1.66 ± 1.48	1.60 ± 0.95	0.74
Jebsen Hand Function Test			
Writing	11.71 ± 2.81	10.72 ± 4.73	0.78
Cards	4.21 ± 2.28	3.97 ± 3.18	0.43
Small	7.86 ± 2.21	8.41 ± 4.82	0.05
Checkers	10.43 ± 3.58	9.69 ± 4.45	0.60
Feeding	11.36 ± 3.00	11.45 ± 3.71	0.32
Light	10.21 ± 3.34	9.66 ± 5.61	0.38
Heavy	9.79 ± 3.27	9.41 ± 4.81	0.45
Perdue Pegboard Test			
Affected hand	8.64 ± 3.07	8.41 ± 6.12	0.40
Both hands	6.86 ± 2.14	5.55 ± 5.21	0.58
Assembly	16.43 ± 8.70	14.34 ± 12.94	0.63
Michigan Hand Outcomes Questionnaire			
Function	19.64 ± 17.16	18.10 ± 13.19	0.83
ADL	21.07 ± 20.38	20.69 ± 16.46	0.91
Work	23.21 ± 22.49	15.52 ± 18.19	0.20
Pain	50.71 ± 19.42	58.45 ± 17.38	0.16
Aesthetics	15.63 ± 19.95	16.16 ± 17.72	0.58
Satisfaction	19.64 ± 19.67	20.69 ± 16.46	0.73

VR, Virtual Reality; CON, conventional; TBSA, total burn surface area; ADL, activities of daily living; Values are presented as mean ± standard deviation, *p*-values were calculated using Fisher’s exact test or a Mann–Whitney test and a Student’s *t*-test.

**Table 2 jcm-09-00731-t002:** The changes of pre-intervention and post-intervention in both groups.

	VR Group (*n* = 28)	CON Group (*n* = 29)	*p*-Value
Grasp and Pinch Power Test			
Grasp (kg)	3.71 ± 4.02	1.78 ± 3.17	** 0.03
Lateral Pinch (kg)	1.31 ± 1.42	−0.46 ± 2.41	** 0.002
Tip Pinch (kg)	0.95 ± 0.88	0.19 ± 0.95	* 0.03
Jebsen Hand Function Test			
Writing	0.64 ± 1.95	0.93 ± 1.87	0.23
Cards	0.07 ± 3.23	0.07 ± 3.13	0.96
Small	2.36 ± 2.45	−0.31 ± 3.24	** <0.001
Checkers	0.57 ± 3.71	0.62 ± 4.32	0.81
Feeding	0.64 ± 2.42	−0.24 ± 2.85	** 0.04
Light	1.00 ± 3.10	1.48 ± 5.47	0.69
Heavy	0.07 ± 3.16	1.14 ± 3.99	0.90
Perdue Pegboard Test			
Affected hand	1.86 ± 2.46	2.21 ± 4.39	0.58
Both hands	2.93 ± 4.40	2.17 ± 3.42	0.69
Assembly	5.86 ± 8.31	5.62 ± 10.54	0.96
Michigan Hand Outcomes Questionnaire			
Function	32.14 ± 11.50	26.72 ± 18.82	0.28
ADL	37.14 ± 15.95	22.59 ± 17.25	** 0.003
Work	26.64 ± 28.25	22.07 ± 17.55	0.37
Pain	−23.21 ± 24.62	−13.62 ± 21.25	0.12
Aesthetics	21.88 ± 24.62	15.22 ± 22.93	0.24
Satisfaction	30.95 ± 10.72	15.95 ± 17.22	** 0.001

VR: Virtual Reality; CON: conventional; * *p* < 0.05 independent *t*-test; measurements between groups were compared; ** *p* < 0.05 Mann–Whitney test; measurements between groups were compared.

**Table 3 jcm-09-00731-t003:** Scores of hand function tests and the Michigan Hand Outcomes Questionnaire after intervention.

	VR Group (*n* = 28)	CON Group (*n* = 29)	*p*-Value
Grasp and Pinch Power Test			
Grasp (kg)	8.06 ± 6.61	6.37 ± 5.91	0.13
Lateral Pinch (kg)	4.54 ± 1.75	3.50 ± 2.69	0.06
Tip Pinch (kg)	2.61 ± 1.83	1.80 ± 1.29	0.07
Jebsen Hand Function Test			
Writing	12.36 ± 1.31	11.66 ± 4.32	0.33
Cards	4.29 ± 3.14	4.03 ± 3.21	0.69
Small	10.21 ± 1.64	8.10 ± 3.49	* 0.01
Checkers	11.00 ± 3.08	10.31 ± 3.01	0.33
Feeding	12.00 ± 2.80	11.21 ± 3.81	0.55
Light	11.21 ± 2.56	11.14 ± 3.60	0.40
Heavy	9.86 ± 3.50	10.55 ± 3.55	0.39
Perdue Pegboard Test			
Affected hand	10.50 ± 3.24	10.62 ± 5.18	0.74
Both hands	9.79 ± 5.88	7.72 ± 4.53	0.33
Assembly	22.29 ± 8.20	19.97 ± 11.72	0.41
Michigan Hand Outcomes Questionnaire			
Function	51.79 ± 17.91	44.83 ± 19.34	0.16
ADL	58.21 ± 18.92	43.28 ± 18.48	* 0.005
Work	47.86 ± 19.88	37.59 ± 13.73	* 0.04
Pain	27.50 ± 16.64	44.83 ± 26.57	* 0.002
Aesthetics	37.50 ± 19.39	31.38 ± 20.17	0.25
Satisfaction	50.60 ± 18.35	36.64 ± 17.55	* 0.02

VR: Virtual Reality; CON: conventional; * *p* < 0.05 Mann–Whitney test; measurements before the intervention and immediately following the 4-week intervention were compared.

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
