# Peer review of "Effects of Virtual Reality-Based Rehabilitation on Burned Hands: A Prospective, Randomized, Single-Blind Study"

_jcm, 2020, doi:10.3390/jcm9030731_

Round 1

Reviewer 1 Report

Thank you for this interesting article.

First of all, this is no article evaluation VR. The authors do not seems to really understand what virtual reality is. Instead it is a special glove connected to a computer system/computer program, that allows specific hand training.

Therefore the authors should delete VR everywhere in the article and change this against a different description. A computer game/program used for hand rehabilitation is interesting but has nothing to do with VR. In this contest some of the references are not quoted correctly.

In detail is would be interesting not only how high the TBSA was, but instead how much TBSA of a hand was injured. This information is not given. Additionally the depth of the hand burn is not described. It is not very important for this publication if the patient had a third degree burn of the trunk. Instead the TBSA of the hand, the burn depth of the injured hand or both hands and the therapy wound be interesting (skin grafting on the hand?). This information is much mure relevant for the clinical outcome of the hand function. 

Additionally it would be interesting if the dominant hand was injured.

How was the range of motion before and after therapy? This data would be very interesting and should be added.

Author Response

  1. First of all, this is no article evaluation VR. The authors do not seems to really understand what virtual reality is. Instead it is a special glove connected to a computer system/computer program, that allows specific hand training. Therefore the authors should delete VR everywhere in the article and change this against a different description. A computer game/program used for hand rehabilitation is interesting but has nothing to do with VR.

Answer> We appreciate you careful advise. We agree that The RAPAEL              Smart GloveTM (Neofect, Yong-in, Korea) is similar with a computer            game/program used for hand rehabilitation. And also the level of realism of the virtual stimuli seems to be low. However, the RAPAEL Smart GloveTM (Neofect, Yong-in, Korea) is a VR rehabilitation tool designed for distal upper extremity. Shin et el. reported that VR-based rehabilitation using RAPAEL Smart GloveTM system had the effectiveness on distal upper extremity function in patients with stroke. We added the references in the introduction section.

  1. In this contest some of the references are not quoted correctly.

Answer> We appreciate you careful advise. We recheck the references and added

descriptions.

  1. In detail is would be interesting not only how high the TBSA was, but instead how much TBSA of a hand was injured. This information is not given. Additionally the depth of the hand burn is not described. It is not very important for this publication if the patient had a third degree burn of the trunk. Instead the TBSA of the hand, the burn depth of the injured hand or both hands and the therapy wound be interesting (skin grafting on the hand?). This information is much mure relevant for the clinical outcome of the hand function. Additionally it would be interesting if the dominant hand was injured.

Answer> We agree with the reviewer. To evaluate the effectiveness of VR rehabilitation accurately, the comparison was limited to the right hand and wrist, and the scars which undergone STSG. We added the descriptions of inclusion criterias in method section.

  1. How was the range of motion before and after therapy? This data would be very interesting and should be added.

Answer> We agree with the reviewer. For detailed information, the hand ROM parameter could be analyzed. However the hand ROM parameter is not included in routine hand function evaluations. We add the limitations of this study in the discussion section.

Reviewer 2 Report

VR is an interesting intervention, and this may be a useful addition to therapy by being more engaging for patients and a good distraction from pain. This may therefore increase time spent (or dose) of therapy, and may result iin better outcomes.  

Your introduction required a clearer rationale for the study, and stronger appraisal and synthesis of the evidence to justify your study.

The font in figure 2 is  too small, making it hard to read. You also need to explain exactly why people dropped out of the study and at what timepoint (i.e. did any of them receive any of the interventions? how many?)

You should also include a TIDIER checklist to give more detail on both interventions including minutes per day for both groups. 

The reasons for the different statistical tests are not explained, and lines 124-138 are very difficult to follow.  I assume you checked data distribution for normality then chose either parametric (t-test) or non-parametric (Wilcoxon) but this is not clear.

Given the number of statistical tests performed on a very small sample, I think there is a high likelihood of a type 1 error in your results, and this needs to be acknowledged in the limitations section.

You would expect improvement in people undergoing 4 weeks of therapy, so I think the within-group pre-post comparisons are unnecessary (i.e. delete tables 2 and 3 and any text that refers to this).

You showed that the sample were similar at baseline, so I would recommend focusing only on the results reported in Table 4. This would make results much easier to interpret. 

It is not clear how you blinded your outcome assessor - more detail is required here.

In your Discussion section, the first paragraph should just summarise your findings (that there was no statistically significant difference between groups for hand strength measures, but the VR group rated themselves significantly better on ADL, work, pain and satisfaction subscales of the MHQ). You then need to comment whether the differences are clinically meaningful, and where possible compare your results to others published for comparable populations (adult burns). The paragraphs from line 207-235 belong in the Introduction section, as they don't relate to your findings at all. 

In your discussion you also need to explain your findings - for example, it may be that people in the VR group received 30 mins more per day of therapy than the control group, so it may be that they just exercised more, or they interacted with their therapists more. Also, how much access did participants have to the VR machine? could they use it any time they wanted? were they in the patient's own room?

Author Response

  1. Your introduction required a clearer rationale for the study, and stronger appraisal and synthesis of the evidence to justify your study.

Answer> We appreciate you careful advise. We added the rationale for this study in introduction section.

  1. The font in figure 2 is too small, making it hard to read.

Answer> We revised the graphic presentations clearer(Figure 2). We hope this will help the reader to understand with ease.

  1. You also need to explain exactly why people dropped out of the study and at what timepoint (i.e. did any of them receive any of the interventions? how many?). You should also include a TIDIER checklist to give more detail on both interventions including minutes per day for both groups. 

Answer> We agree with the reviewer. We add the additional text in the method section.

  1. The reasons for the different statistical tests are not explained, and lines 124-138 are very difficult to follow. I assume you checked data distribution for normality then chose either parametric (t-test) or non-parametric (Wilcoxon) but this is not clear.

Answer> We appreciate you careful advise. We agree with the reviewer. So the statistical explanations were added in materials and methods section.

  1. Given the number of statistical tests performed on a very small sample, I think there is a high likelihood of a type 1 error in your results, and this needs to be acknowledged in the limitations section.

Answer> We agree with the reviewer. We add the additional text in the limitations section.

  1. You would expect improvement in people undergoing 4 weeks of therapy, so I think the within-group pre-post comparisons are unnecessary (i.e. delete tables 2 and 3 and any text that refers to this). You showed that the sample were similar at baseline, so I would recommend focusing only on the results reported in Table 4. This would make results much easier to interpret. 

Answer> We agree with the reviewer. We revised the descriptions, which are focusing more the Table 4, in the result section..

  1. It is not clear how you blinded your outcome assessor - more detail is required here.

Answer> We agree with the reviewer. We revised the materials and method section as your advise.

  1. In your Discussion section, the first paragraph should just summarise your findings (that there was no statistically significant difference between groups for hand strength measures, but the VR group rated themselves significantly better on ADL, work, pain and satisfaction subscales of the MHQ). You then need to comment whether the differences are clinically meaningful, and where possible compare your results to others published for comparable populations (adult burns).

Answer> We agree with the reviewer. We added the descriptions in discussion section and added more references related with the this study.

  1. The paragraphs from line 207-235 belong in the Introduction section, as they don't relate to your findings at all. 

Answer> We appreciate you careful advise. We revised the discussion section. We hope this will help the reader to understand with ease.

  1. In your discussion you also need to explain your findings - for example, it may be that people in the VR group received 30 mins more per day of therapy than the control group, so it may be that they just exercised more, or they interacted with their therapists more. Also, how much access did participants have to the VR machine? could they use it any time they wanted? were they in the patient's own room?

Answer> We agree with the reviewer. We add the additional text in the method section.

Round 2

Reviewer 1 Report

Thank you for resubmitting the manuscript. All my questions have been answered.

Author Response

We appreciate you careful advise.

Reviewer 2 Report

See edits to manuscript file.

Also, it is unclear if this study had approval from a Human Research Ethics board. The authors state that the study was approved by ClinicalTrials, but that is NOT approval, just registration. Please include ethics approval number and the committee that approved it.

There was also no pre-trial calculation for statistical power (to determine how many people they needed to enrol) - if this was done, it should be included.

In terms of results, I think the authors have misunderstood my original recommendation as they have still included within-group comparisons in results text (and tables 2 and 3) which are unnecessary and confusing to readers. Just include a comparison of control vs intervention scores (or change scores, which is final score minus baseline score).

Also, comment on whether differences between groups were clinically significant using any data published on the Minimum Clinically Important Change for each instrument.

Author Response

  1. Also, it is unclear if this study had approval from a Human Research Ethics board. The authors state that the study was approved by ClinicalTrials, but that is NOT approval, just registration. Please include ethics approval number and the committee that approved it.

Answer> We agree with the reviewer. We add the ethics approval number and the committee that approved it in the method section.

  1. There was also no pre-trial calculation for statistical power (to determine how many people they needed to enrol) - if this was done, it should be included.

Answer> We agree with the reviewer. Shin et el. reported that VR-based rehabilitation using RAPAEL Smart GloveTM system had the effectiveness on distal upper extremity function in patients with stroke. We added the references and descriptions in the method section.

  1. In terms of results, I think the authors have misunderstood my original recommendation as they have still included within-group comparisons in results text (and tables 2 and 3) which are unnecessary and confusing to readers. Just include a comparison of control vs intervention scores (or change scores, which is final score minus baseline score).

Answer> We agree with the reviewer. We eliminate tables 2 and 3, and added the new table 2 for comparing the change scores. We revised the method section and the result section. We hope this will help the reader to understand with ease.

  1. Also, comment on whether differences between groups were clinically significant using any data published on the Minimum Clinically Important Change for each instrument.
  2. Answer> We agree with the reviewer. Mimimum clinically important changes for scoring system in the JTT and MHQ have not been established in patients with burns. We added the descriptions and references in the discussion section.